# Establishing Breeding Priorities for Developing Biofortified High-Yielding Pearl Millet (*Pennisetum glaucum* (L.) R. Br.) Varieties and Hybrids in Dosso Region of Niger

Bassirou Sani Boubacar Gaoh [1,2], Prakash I Gangashetty [1,3,*], Riyazaddin Mohammed [1],
Mahalingam Govindaraj [3,4], Daniel Kwadjo Dzidzienyo [2] and Pangirayi Tongoona [2]

1　International Crops Research Institute for the Semi-Arid Tropics (ICRISAT), Niamey BP 12404, Niger
2　West African Centre for Crop Improvement, College of Basic and Applied Sciences, School of Agriculture, University of Ghana, PMB LG 30 Legon, Ghana
3　International Crops Research Institute for the Semi-Arid Tropics (ICRISAT), Patancheru 502324, India
4　HarvestPlus, Alliance of Bioversity International and the International Center for Tropical Agriculture (CIAT), Cali 760001, Colombia
*　Correspondence: p.gangashetty@cgiar.org

**Abstract:** West Africa is the origin and epicenter of pearl millet genetic diversity. Niger is a standalone country that produces 3.5 million tons of pearl millet from an area of 6.7 million hectares, with productivity varying from 0.5 to 0.7 t/ha. Low grain yield is a result of low soil fertility, drought, downy mildew, head miner, and the non-utilization of improved and quality seeds. Around 30 pearl millet varieties were released in Niger, but the adoption rate of improved varieties is still lagging. There has been no systematic mapping implemented for new varieties' adoption preferences and the availability of quality seeds. Considering this and assessing the need for biofortified cultivars, the present participatory study was conducted in the Dosso region of Niger, wherein high rates of malnutrition persist. This study aimed (i) to identify breeding priorities for key traits of pearl millet preferred by farmers, with gender-based segregation, for varieties and hybrids, and (ii) to survey the preference for biofortified varieties with added nutritional value. Structured questionnaires and focus groups were used to collect data from 150 randomly selected respondents in 12 villages from three representative departments of Dogondoutchi, Dosso, and Gaya. The results reveal that pearl millet is a primary staple crop grown (98% of respondents) and consumed on a daily basis as food and also used as feed for their animals. The majority of farmers preferred a long panicle (50.7%) and a good seed set (45.3%). For grain traits, a white color (50%) and larger size (100%) were predominantly preferred, which fetches them higher prices in the market, where they compete with sorghum grains. All respondents unanimously rated growing biofortified pearl millet varieties as high (100%), owing to higher Fe and Zn, in addition to yield. Furthermore, 99.3% of farmers perform grain decortication before consumption, thus potentially depleting staple grain nutrition, which is expedient for pearl millet biofortification in the region. This study has the potential for establishing pearl millet breeding priorities that are likely to be employed for other West African pearl millet breeding programs.

**Keywords:** biofortification; iron; zinc; pearl millet; West Arica

## 1. Introduction

Pearl millet (*Pennisetum glaucum* (L.) R. Br.), a climate-resilient cereal, is an important food crop in the dry regions of the Sahel. The majority of the cultivated area of pearl millet is in Africa (~18.0 m ha), Asia (~10.0 m ha), and the Americas (~2.0 m ha) [1]. In Niger, pearl millet production has reached 3.5 m tons, making Niger the largest pearl-millet-producing country in Africa [2]. This crop accounts for three-quarters of the cereal production and over half the cultivated land in Niger [3], and the highest per capita consumption of millet is in Niger [4].

Pearl millet grains have higher levels of grain Fe and Zn contents when compared to wheat and rice [5]. The energy density of pearl millet is also relatively high, arising from its higher oil content relative to maize, wheat, and sorghum [6]. Despite these properties and being the central component of food security in dry areas [7], people in Niger are affected by poor-quality diets, resulting in micronutrient malnutrition from Fe and Zn deficiency. To address malnutrition, biofortification has been described as a cost-effective (only requires a one-time investment) and sustainable strategy that provides a long-term positive effect [8]. Through HarvestPlus, which is part of the CGIAR Research Program, many biofortified varieties have been released to combat hidden hunger, i.e., beans with Fe, pearl millet with Fe, cassava with vitamin A, maize with vitamin A and Zn, sweet potato with vitamin A, rice and wheat with Zn [9]. In Niger, the improved high-iron pearl millet variety "Chakti" was released in 2018, and efforts are underway to scale up Africa's first Fe-biofortified pearl millet variety. Moreover, pearl millet's most widely cultivated variety HKP in Niger was found to have low grain Fe and Zn contents compared to the biofortified varieties.

Although several high-yielding varieties were released in Niger (around 30 improved varieties) [10], their adoption has been very poor because of the underdeveloped and undertrained seed systems in this region [11]. According to the 2019 annual yearbook only eight improved varieties were available for the rainy season in 2019. The HKP variety alone accounted for 90% of improved seeds available, followed by SOSAT-C88 (6%), whereas other improved varieties accounted for only 4% [12]. It was reported that less than 2% of the total national seeds planted by farmers are supplied by formal seed systems [13]. Hence, the focus should be placed on seed systems using seed-based projects, while varietal development needs to be accelerated using participatory approaches to engage farmers and in utilizing the quality seeds of improved varieties and hybrids.

There is strong confirmation for the hypothesis that the insufficient priority given to consumer-preferred traits by breeding programs contributes to the limited uptake of modern varieties and low varietal turnover [14]. In Sub-Saharan Africa, farmers are most often the consumers of the food crops they grow, and they, therefore, have developed particular preferences for the crops they grow to match their food types and needs. Buerkert et al. reported a preference for pearl millet varieties with long panicles in the Sahel, as they provide an important advantage for Sahelian farmers whose mode of transport (on camel or donkey backs) and commerce are based on millet bundles (botes) that contain up to 200 similar long panicles tied together [15]. For grain shape preference, the classification of the pearl millet species of the world based on seed shape indicated that the Globosum race was the most common race found in central Nigeria, Niger, Ghana, Togo, and Benin [16].

To our knowledge, no investigations concerning the adoption and acceptance of released and pipeline-breeding high-iron pearl millet varieties have been conducted in Niger. Therefore, this participatory rural appraisal (PRA) study was conducted to understand whether farmers in Niger would prefer varieties with added nutritional value, along with the gender-based trait priorities for breeding. In addition, other attributes that farmers prefer and the trade-offs between the nutritional and other important production/consumption attributes were evaluated. The study also aimed to segregate and record the trait preferences on gender basis.

## 2. Materials and Methods

### 2.1. Study Area

The study area was in the Republic of Niger; regions are first-level subdivisions (7 regions), while departments are second-level subdivisions (63 departments). The present study was carried out in three departments Dogondoutchi, Dosso, and Gaya of the Dosso region, located in the southern part of Niger (Figure 1). In Niger, pearl millet activities are carried out in the rainy season (June to September), which has a unimodal pattern. The Dogondoutchi, Dosso, and Gaya departments are located in the northern, central, and southern parts of the Dosso region and receive an average annual rainfall of 456, 564, and

811 mm, respectively [17]. The participatory rural appraisal (PRA) was carried out in four villages from each department in 2017 (Supplementary Table S1).

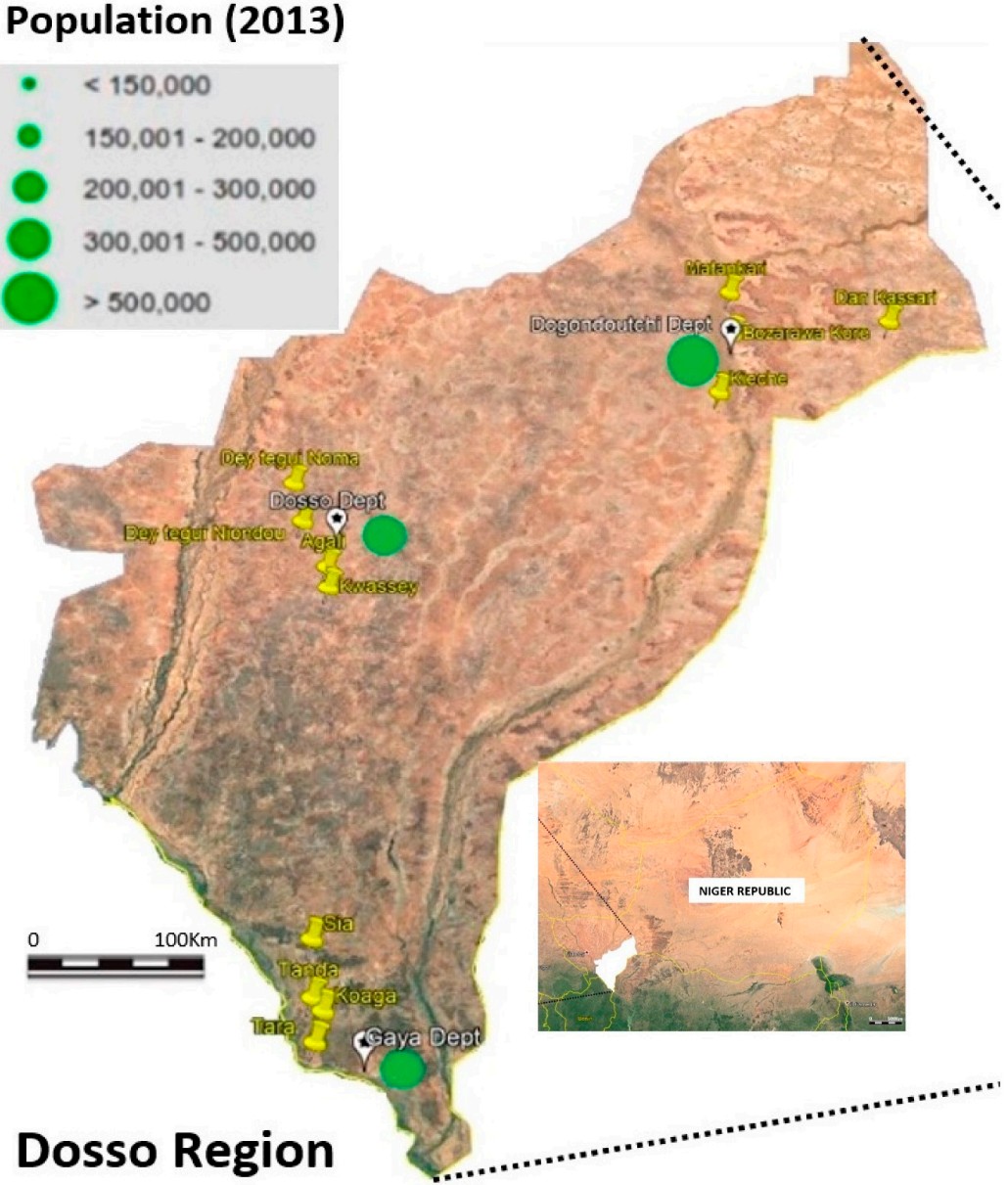

**Figure 1.** Departments of the Dosso region in Niger, in yellow are the villages where a participatory rural appraisal (PRA) was conducted, Black pentagram represents the departments where the study was conucted (modified from Google Earth 2018).

### 2.2. Selection of Study Sites and Sampling Method

The study site selection and the sampling method were based on a multistage sampling method to select the respondents. First, the Dosso region in Niger was selected based on reports from the National Institute of Statistics, which stated that Dosso was the most micronutrient-deficient region in Niger [17]. Second, a purposive approach was used to select three departments Gaya, Dosso, and Dogondoutchi based on their contrasting ecology due to differences in the average annual rainfall and socio-economic environments. The third consisted of convenience sampling in four villages from each department that were easily accessible by roads. The fourth consisted of the purposive selection of farmers by local extension agents, followed by the random selection of 5 to 20 respondents from each village.

A reconnaissance survey was performed through short visits with an extension agent of the villages, where focus group discussions (FGDs) took place and ideas were exchanged with key informants (village leaders) on issues that could affect planning. In addition, transect walks were carried out to gain information on the local environment.

### 2.3. Data Collection and Analysis

The data were collected through PRA techniques using FGDs and individual interviews (see Supplementary Materials for the questionnaire). In each department, two FGDs took place in each village, with men (pearl millet smallholders) and women surveyed separately to avoid biases. The extension agents were the PRA facilitators in each region. The FGD started with an introduction followed by a mission statement so that interaction would be easier. A checklist of topics to be covered was prepared in advance. The main topics were as follows. (A) Crops cultivated by the interviewees of the community. (B) Local perception of malnutrition. (C) How many different varieties of millet were grown? (D) Varietal traits preferred by farmers. (E) Pearl millet production constraints. A semi-structured questionnaire was used for individual interviews to ascertain and supplement the FGD findings. The main topics were pearl millet farming practices, varietal and grain preferences, production constraints, dietary diversity, and grain nutritional preference by men and women farmers. All interviewed farmers estimated their yield by counting the number of harvested pearl millet bundles. One bundle of pearl millet is estimated to be equivalent to 15 kg of grains [18]. The pearl millet grain yield was computed after determining the quantity of the pearl millet grain harvested and estimating the acres of land under pearl millet cultivation. The Census and Survey Processing System software (CSPro 7.0.2 of 29 June 2017) was used to collect and organize data. The generated data were analyzed using the Statistical Package for Social Sciences (SPSS 25). Variables were subjected to frequency distribution analysis.

## 3. Results

### 3.1. Pearl Millet Importance and Farming Practices

Pearl millet was ranked as the most important crop grown by farmers in the villages where the FGDs were conducted, except in Tara village, where women consider sorghum to be the most important crop, followed by pearl millet, and in Deytagui Noma village, where women do not grow pearl millet and consider Bambara groundnut the most important crop (Table 1). Cowpea was ranked second in Deytagui Noma and Bozawara Kore villages, while rice was ranked second after sorghum in Tara.

**Table 1.** Cultivated crops ranked according to importance to farmers in three departments (Dogondoutchi, Dosso, and Gaya) of the Dosso region in Niger Republic.

| | Dosso Region | | | | | |
| | Dosso Department | | Gaya Department | | Dogondoutchi Department | |
| Crop | Deytagui Noma | | Tara | | Bozawara Kore | |
| | Men | Women | Men | Women | Men | Women |
|---|---|---|---|---|---|---|
| Pearl millet | 1 | - | 1 | 2 | 1 | 1 |
| Sorghum | 6 | - | 3 | 1 | 4 | 7 |
| Bambara groundnut | 4 | 1 | - | 6 | 5 | 2 |
| Cowpea | 2 | 4 | 4 | - | 2 | 4 |
| Corn | 10 | 2 | 6 | 4 | 6 | 8 |
| Groundnut | 5 | 3 | 5 | 5 | 3 | 3 |

**Table 1.** *Cont.*

| Crop | Dosso Region | | | | | |
| | Dosso Department | | Gaya Department | | Dogondoutchi Department | |
| | Deytagui Noma | | Tara | | Bozawara Kore | |
| | **Men** | **Women** | **Men** | **Women** | **Men** | **Women** |
| Cucurbitacee gourgi | - | - | - | - | - | 11 |
| Cucurbita maxima | 8 | - | - | - | - | - |
| Lettuce | - | - | 10 | 9 | - | - |
| Moringa oleifera | - | 9 | - | - | - | 12 |
| Okra | 7 | 7 | - | - | - | 6 |
| Onion | - | - | 9 | - | - | - |
| Cabbage | - | - | 8 | 8 | - | - |
| Pepper | - | - | - | 13 | - | 13 |
| Rice | - | - | 2 | 3 | - | - |
| Roselle | 3 | 5 | - | 7 | - | 9 |
| Sesame | 9 | 6 | 7 | 14 | - | 5 |
| Cassava | - | - | - | 10 | - | - |
| Squash | - | 8 | - | - | - | 10 |
| Sweet potato | - | - | - | 12 | - | - |
| Tomato | - | - | - | 11 | - | - |

Pearl millet production is exclusively carried out in the rainy season and intercropped with other crops, and the majority of farmers (97.3%) have been practicing pearl millet farming since their childhood (Table 2). Most of the farmers treated the seeds (83.3%) before sowing and performed thinning (98.7%). The average number of plants per hill was most often 3 to 4 (68.7%). The farmers mostly used a combination of manure and fertilizer (61.3%) and used their footsteps (approximately 76 cm) as a reference during sowing for plant spacing (95.3%). The majority of farmers produced pearl millet on the same land every year (75.3%), while 24.7% combined pearl millet farming with fallowing.

The farmers mostly grow two pearl millet varieties (71.3%) during the rainy season: early- and late-maturing varieties. They obtained seeds from their own stocks (98%) and did not purchase pearl millet seeds (83.1%) (Table 2). The pearl millet landraces grown by a high proportion of the farmers (69.33%) have a low grain yield of between 0.3 and 0.6 t/ha.

**Table 2.** Pearl millet farming practices in three departments (Dogondoutchi, Dosso, and Gaya) of the Dosso region in Niger.

| Question | Scale | Percentage (%) |
| --- | --- | --- |
| How long have you been cultivating pearl millet? | Since childhood | 97.3 |
| | Adult | 2.7 |
| When do you plant pearl millet? | Rainy season | 100.0 |
| | Off season | 0.0 |
| Do you treat the seeds? | Yes | 83.3 |
| | No | 16.7 |

**Table 2.** *Cont.*

| Question | Scale | Percentage (%) |
|---|---|---|
| Do you perform thinning? | Yes | 98.7 |
| | No | 1.3 |
| What is the average number of plants per hill? | 3–4 | 68.7 |
| | 5–6 | 28.0 |
| | 7–8 | 3.3 |
| Do you apply fertilizers? | Organic | 35.3 |
| | Mineral | 0.7 |
| | Both | 61.3 |
| | None | 2.7 |
| What is the spacing between plants? | 80 cm | 0.7 |
| | 1.00 m | 4.0 |
| | 1 Footstep | 95.3 |
| Do you perform fallow? | Yes | 24.7 |
| | No | 75.3 |
| Do you grow other crops with pearl millet? | Yes | 100.0 |
| | No | 0.0 |
| How many different varieties do you grow? | 1 | 26.7 |
| | 2 | 71.3 |
| | 3 | 2.0 |
| Do you purchase pearl millet seeds often? | Yes | 18.7 |
| | No | 81.3 |
| Where do you obtain pearl millet seeds? | Own stock | 98.0 |
| | Market | 2.0 |
| What are the yields (t/ha) of the pearl millet that you grow? | 0.15–0.26 | 12.7 |
| | 0.30–0.60 | 69.3 |
| | 0.62–1.00 | 18.0 |

*3.2. Pearl Millet Varietal and Grain Preferences*

Earliness and stalk tillering were the most preferred traits by men and women during FGDs in Deytigui Noma and Bozawara Kore villages, while men in Tara considered adaptation and earliness to be the most important traits (Table 3). Women considered a bigger grain size and earliness in Deytagui Noma, grain yield in Tara, and stalk tillering in Bozawara Kore to be their top priorities.

All respondents were willing to purchase and grow improved pearl millet varieties (Table 4). For grain characteristics, the most preferred traits were a white grain color (50%), big grain size (100%), and longer panicle (59.3%). A major proportion of farmers (93.3%) do not consider grain shape to be an important trait.

**Table 3.** Farmers' pearl millet trait preferences ranked according to importance in three departments (Dogondoutchi, Dosso, and Gaya) of the Dosso region in Niger.

| | Dosso Region | | | | | |
| --- | --- | --- | --- | --- | --- | --- |
| | Dosso Department | | Gaya Department | | Dogondoutchi Department | |
| | Deytagui Noma | | Tara | | Bozawara Kore | |
| | **Men** | **Women** | **Men** | **Women** | **Men** | **Women** |
| Earliness | 1 | 2 | 2 | 3 | 1 | 3 |
| Yield | 4 | - | 3 | 1 | 4 | 4 |
| Long Panicle | 5 | - | 4 | 2 | 3 | 2 |
| Tillering | 2 | - | - | - | 2 | 1 |
| Nutritional | 3 | 3 | - | - | 5 | - |
| Bigger Seed | 6 | 1 | 5 | - | - | - |
| Adaptation | - | - | 1 | 4 | - | - |

**Table 4.** Farmers' pearl millet varietal and grain preferences in three departments (Dogondoutchi, Dosso, and Gaya) of the Dosso region in Niger.

| Question | Response | Percentage (%) |
| --- | --- | --- |
| Do you know any improved millet varieties? | Yes | 65.3 |
| | No | 34.7 |
| Are you willing to try improved pearl millet varieties? | Yes | 100.0 |
| | No | 0.0 |
| Will you purchase seeds every year if available? | Yes | 100.0 |
| | No | 0.0 |
| Which kind of pearl millet grain color do you prefer? | White | 50.0 |
| | Red | 28.7 |
| | Black | 16.0 |
| | Not important | 5.3 |
| Which kind of pearl millet grain size do you prefer? | Big | 100.0 |
| | Medium | 0.0 |
| | Small | 0.0 |
| | Not important | 0.0 |
| Which kind of pearl millet grain shape do you prefer? | Globular | 6.7 |
| | Not important | 93.3 |
| What kind of panicle do you prefer? | Small (ICTP type) | 4.7 |
| | Medium (Tabi type) | 35.3 |
| | Longer (Zongo type head) | 59.3 |
| | No preference | 0.7 |
| What are your preferred traits in pearl millet? | Seed set | 45.3 |
| | Earliness | 2.0 |
| | Tillers more | 0.7 |
| | Long panicle | 50.7 |
| | Good stand | 1.3 |

### 3.3. Pearl Millet Production Constraints

The FGDs revealed that the most important pearl millet production constraint was poor soil fertility, except for women in Tara, who considered pearl millet flower pests (*Dysdercus superstitiosus*, *Rhinyptia infuscata* Burm., and *Decapotoma affinis* Billb.) to be more important (Table 5). In Bozawara Kore, the second most important constraint was drought, while in Deytagui Noma and Tara, men considered *Striga hermonthica* the second most important constraint on pearl millet production.

**Table 5.** Farmers' pearl millet production constraints ranked according to importance in three departments (Dogondoutchi, Dosso, and Gaya) of the Dosso region in Niger.

| Constraint's | Dosso Region | | | | | |
|---|---|---|---|---|---|---|
| | Dosso Department | | Gaya Department | | Dogondoutchi Department | |
| | Deytagui Noma | | Tara | | Bozawara Kore | |
| | Men | Women | Men | Women | Men | Women |
| Poor soil fertility | 1 | - | 1 | 4 | 1 | 1 |
| Drought | - | - | - | - | 2 | 2 |
| *Coniesta ignefusalis* | 3 | - | - | - | - | - |
| Dysdercus | - | - | 3 | 1 | 3 | 3 |
| Striga | 2 | - | 2 | 3 | - | - |
| Weeds | - | - | - | 2 | - | - |
| Locust | - | - | - | - | - | 4 |

Forty-eight percent of the surveyed farmers reported poor soil fertility as being the most important pearl millet production constraints, followed by drought (24%) and *Striga hermonthica* (19%) in the Dosso region (Figure 2). Pest damage was reported as a variable each year and was considered less important (8%).

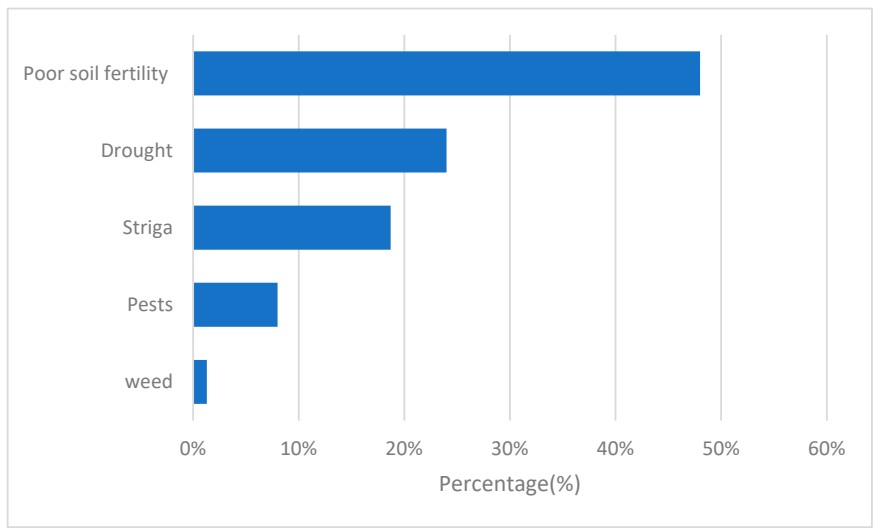

**Figure 2.** Pearl millet production constraints in three departments (Dogondoutchi, Dosso, and Gaya) of the Dosso region in Niger.

### 3.4. Local Perception of Malnutrition and Dietary Diversity

The FGDs in the three departments revealed a big disparity between men and women regarding malnutrition. Men attributed hidden hunger to the lack of satiation, while women were more aware of micronutrient deficiencies and their consequences, mostly

anemia. All interviewed farmers indicated their interest in growing millet rich in Fe and Zn for their children and lactating women (Table 6).

**Table 6.** Pearl millet consumption in three departments (Dogondoutchi, Dosso, and Gaya) of the Dosso region in Niger.

| Question | Response/Range | Percentage (%) |
|---|---|---|
| What quantity of pearl millet do you eat weekly? | 0.84 kg to 3.5 kg | 26.7 |
| | 3.57 kg to 7 kg | 54.0 |
| | 7.49 kg to 11.69 kg | 19.3 |
| Do you always perform decortication of grains for food? | Yes | 99.3 |
| | No | 0.7 |
| Are you interested in growing millet with good nutrition, e.g., Fe and Zn for your kids and lactating women? | Yes | 100.0 |
| | No | 0.0 |

Almost all interviewed farmers (99.3%) performed the decortication of pearl millet grains for food. The survey revealed that, across departments, 98%, 22%, and 14% of farmers consumed pearl millet, meat, and dairy products on a daily basis, respectively (Figure 3). Meat and dairy products were more often consumed in the Dogondoutchi department, whereas they were less often consumed in the Gaya department.

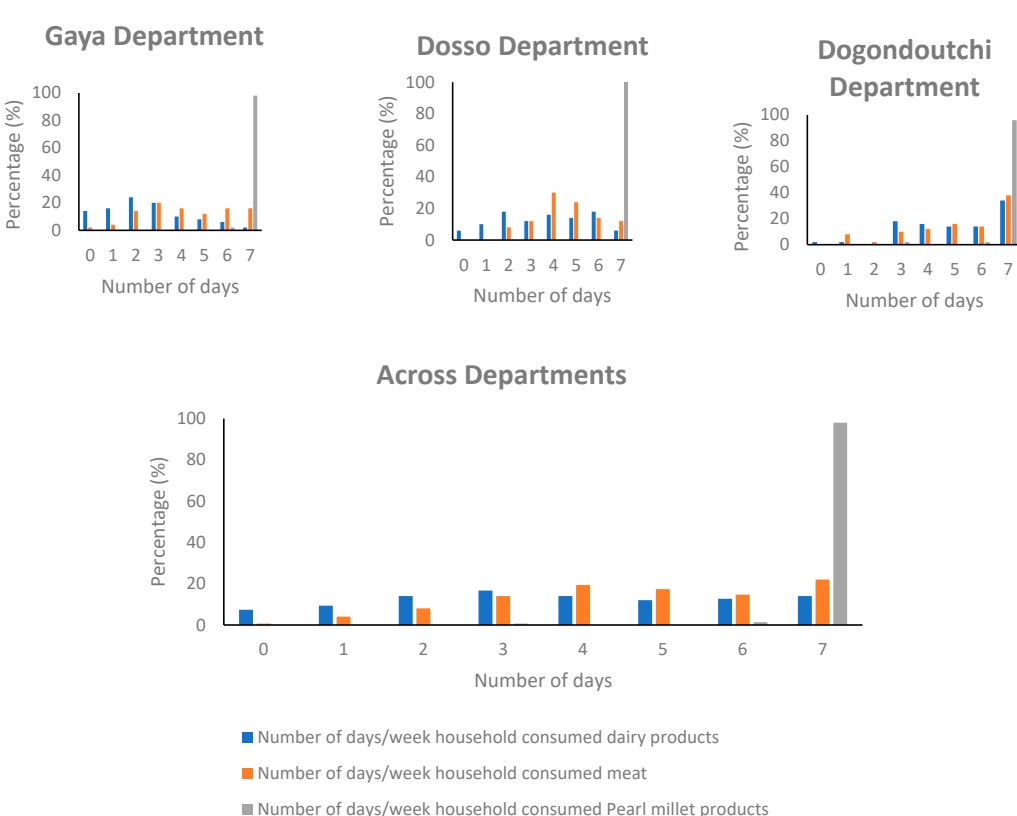

**Figure 3.** Household dietary diversity in three departments (Dogondoutchi, Dosso, and Gaya) of the Dosso region in Niger.

## 4. Discussion

Considering the semi-arid farming practices and food system, pearl millet plays an important role in food and nutrition security in the Sahel region. This is well reflected in the results of FGDs in the Dosso region of Niger. In the Dosso region, pearl millet

cultivation is dominated by men. During FGDs, women in Deytagui Noma stated that when the patriarch was absent during traveling, they were not allowed to sow pearl millet during the rainy season. Additionally, in Tara and Bozawara Kore, women disclosed that they participated in farming activities but did not own the land or decide on cropping. Kanfany et al. (unpublished) also stated similar findings for the groundnut basin in Senegal, where men dominate crop production activities, while women provide operational support.

Interestingly, all farmers in the Dosso region grow two types of pearl millet varieties (late- and early-maturing); they stated that at the beginning of the rainy season, their pearl millet grain stocks are depleted, and an early variety is the only option to provide food grains for their consumption, while the late-maturing variety serves as the major part of their pearl millet production, consumption, and markets to some extent. A similar pattern of practices was reported in Ghana, where pearl millet was harvested at the peak of the rainy season when the farmers' stocks of food grains from the previous harvest were exhausted [19]. The whole farming community (98%) used pearl millet seeds from their own stocks by selecting the best-looking panicles for planting during the following rainy season. Therefore, in this region, farmers underutilized the adaption of improved varieties. However, farmers' selections of trait preferences and maturity within varietal bulks are very critical. In fact, artificial selection coupled with natural selection practiced in pearl millet has led to the development of well-adapted local landraces in the semi-arid region of Africa [20].

FGDs revealed that low grain yields are characteristic of most of the farmers' landraces that produce close to 0.5 t/ha. These findings are in line with Guengant and Banoin, who reported an average pearl millet grain yield of 400 kg/ha in Niger for five decades (1955 to 1999) [21]. Around 96.6% of the farmers reported using largely organic fertilization or a combination of organic and mineral fertilization but were reaping low yields per unit area using local or traditional landraces (haini kiré, darancoba, zongo, omno, etc.), which highlights the need to replace landraces with improved and high-yielding varieties from the ICRISAT and NARS breeding centers.

Regarding the pearl millet varietal and grain preferences, the results revealed that earliness was the most preferred trait by both men and women farmers irrespective of the department. The survey also showed that a good seed set (seed covered > 80% of panicle) combined with a long panicle (>50 cm long) accounted for 96% of the preferred traits. In breeding, both the seed set and long panicle are correlated with the yield per unit area, which suggests that the farmers will likely adopt an early variety with a long panicle and good seed set for better yield [22,23]. Most of the farmers had heard of improved varieties and reported having received seeds of improved varieties. However, these were lost after a few years, probably due to the millet seeds produced by the outcrossing (>80%) reproductive mechanism facilitated by their protogynous nature (the stigma emerges first). There were opportunities to mix seeds with landraces in the household since practicing farmers selected seeds from their own stock. All respondents were willing to purchase improved pearl millet varieties if available in a timely manner in local shops or from suppliers irrespective of the seed cost, which suggests that improving the supply chain of improved pearl millet seed will ensure the seed replacement rate in pearl millet and thereby promote higher regional production to cope with household food and nutrition security.

For pearl millet production constraints, apart from the cultivar choice, the FGDs and questionnaire survey revealed that poor soil fertility was the most important constraint on pearl millet production in the Dosso region. Soil fertility is a major constraint for countries across west and central Africa, where semi-arid to arid conditions historically exist. Earlier studies reported that poor soil fertility and the limited use of organic and mineral fertilizers were the most important constraints on increasing agricultural productivity in West Africa [24]. In West Africa, farmers traditionally used long fallow periods to restore soil fertility. However, less land is available due to the growing population, making these practices ineffective [25].

The second most important constraint was drought. Pearl millet production is mostly practiced on marginal lands with scanty rainfall and drought stress, which represents

a major constraint on the grain yield in these farming systems [26]. Efforts have been made to increase the pearl millet yield through mineral fertilization and the selection of early-maturing varieties that escape late-season drought [27]. Early-maturing pearl millet varieties have a low water requirement, which enables relative drought escape, mostly at the end of the season [28]. In addition, drought stress was found to cause a reduction in the pearl millet seed size. This reduction was mainly due to the reduction in the endosperm component of the grain, making the grain layers a major proportion of the seeds [27]. Annual precipitation can be very low (373.8 mm) and occur in a short span (37 days), which is a big constraint that requires the strengthening of natural and local water reservoirs [17]. This will drive improvements in soil fertility by growing multiple crops or drought-tolerant trees that provide organic matter to fields and support integrated farming.

The parasitic weed *Striga hermonthica* was the third most important constraint on pearl millet cultivation in this study. The farmers reported that they mostly observe striga in farms that have been used for cultivation for a long period. This is consistent with Vogt et al., who reported that striga preferentially occurs in nutrient-poor soils that have been exhausted by continuous cropping [29].

Due to the local perception of malnutrition and dietary diversity, in Deytagui Noma, women suffering from anemia visit the hospital to receive fortified biscuits. Elderly women reported that, during their youth, they used to prepare special porridge for new mothers and their babies. In Tara and Bozawara Kore, women still prepare porridge for new mothers. In Tara, porridge is prepared from whole pearl millet grain, sorghum, sesame, groundnut, and dried fish. Meanwhile, in Bozawara Kore, they use sesame, roselle, squash, and cowpea to prepare porridge for breastfeeding mothers. Across all sites, men associated malnutrition with a lack of satiating hunger, while women at all sites were aware of hidden hunger, especially anemia. Anemia is referred to locally as a "lack of blood", a term used by nurses to explain anemia to new mothers. The expression "lack of blood" refers to the insufficient number of red blood cells or their incapacity to bind oxygen. Iron has many key functions due to its role as an electron donor/acceptor in the human body and is a core part of oxygen transport through the heme complex, which is found in oxygen-binding hemoglobin molecules in red blood cells [30]. To produce hemoglobin, Fe is a major required constituent; it is proven that the consumption of biofortified pearl millet cultivars will provide more than 60% of the daily Fe requirement [5,31].

For instance, the majority of the farmers consumed pearl millet on a daily basis (98%) and consumed at least 3.57 kg of pearl millet weekly, which might suggest that increasing the density of micronutrients, especially iron and zinc in pearl millet grain could reduce iron deficiencies in the Dosso region. Meat and dairy products were more often consumed in the Dogondoutchi department, whereas they were less often consumed in the Gaya department, mostly because, among the three departments, livestock production is the highest in the Dogondoutchi department and the lowest in the Gaya department [17]. All survey respondents were willing to try improved biofortified pearl millet for their children and breastfeeding women. In addition, according to Welch, increasing the quantities of micronutrients stored in seeds can result in improved seedling vigor and viability when these seeds are sown in micronutrient-poor soils, thus enhancing the performance of seedlings during crop establishment [32]. However, 99.3% of the surveyed farmers reported practicing the decortication of pearl millet grain before consumption. Decortication is commonly performed with abrasive disks in mechanical dehullers. Pearl millet grain is usually decorticated of the bran (pericarp) from the endosperm before being consumed in Niger [15]. In a study conducted by Hama et al., they found that half of the iron content was removed when 10% of the pearl millet dry matter was abraded [33]. In fact, the removal of the bran leads to a decrease in the nutrients, including minerals and anti-nutrients, as these are mostly situated in the peripheral area of the grains (pericarp and aleurone layers).

## 5. Conclusions

The present study confirmed the importance of pearl millet as a staple food that contributes both energy and nutrition in the Dosso region. This is evidenced by the predominance of its daily consumption. Farmers grew traditional low-yielding pearl millet landraces and inadequately fertilized their fields, suggesting the need for a map-based soil nutrition guide in the region. A long panicle, high seed set, earliness, bigger grains, and white grains were the most important traits according to farmers. The most important constraint on pearl millet production in the Dosso region was poor soil, followed by drought and the parasitic weed Striga. These can be addressed by holistic breeding objectives and following good agronomic practices. Women were more aware of malnutrition and its consequences than men, implying that the selection of cultivars by women farmers is likely to improve nutrition in households. A large majority of farmers performed grain decortication, which can reduce the grain micronutrient density. Hence, biofortified cultivars with enriched total grain micronutrients are recommended to improve the nutrition and health of resource-poor micronutrient-deficient households in the Dosso region and across Niger. The shortcoming of this study is that the survey was limited to only the region of Niger, and therefore, it will be appropriate to extend future research to the other seven regions of Niger. Finally, it is important to integrate biofortification into Government policies and programs by, for instance, including biofortified varieties in the crop seeds that are distributed to farmers at the beginning of the rainy season.

**Supplementary Materials:** The following supporting information can be downloaded at: https://www.mdpi.com/article/10.3390/agronomy13010166/s1, Table S1: Participatory rural appraisal (PRA) study sites, localization, and the number of farmers interviewed in the Dosso region of Niger.

**Author Contributions:** B.S.B.G. and P.I.G. conceived the study, performed the research and data analysis, and wrote the manuscript with input from D.K.D., R.M. and P.T.; M.G. edited and revised the manuscript. All authors have read and agreed to the published version of the manuscript.

**Funding:** The present research was carried out with the financial support of the German Academic Exchange Service (DAAD) and ECONET (Strive Masiyiwa) funding.

**Acknowledgments:** The authors thank colleagues from the International Crops Research Institute for the Semi-Arid Tropics (ICRISAT), Sahelian Center, Sadoré-Niger. Thanks to Esther Njuguna-Mungai (ICRISAT-Nairobi) and Jummai Yila (ICRISAT-Mali). Thanks to the regional directors and extension agents of agriculture in the Dosso, Gaya, and Dogondoutchi departments. Additionally, thanks to the village chiefs for allowing us to conduct the study in their villages and the farmers who participated in the study. PGS and MG have been supported by HarvestPlus projects.

**Conflicts of Interest:** All authors have carefully read the manuscript, and they do not have any conflicts of interest.

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
