# Peer review of "Establishing Breeding Priorities for Developing Biofortified High-Yielding Pearl Millet (Pennisetum glaucum (L.) R. Br.) Varieties and Hybrids in Dosso Region of Niger"

_agronomy, doi:10.3390/agronomy13010166_

Round 1
Reviewer 1 Report
The article is devoted to determining the priorities of African millet breeding based on a sociological survey of farmers in the Dosso Niger region. The research carried out is very interesting.
However, the article can be published only after careful revision.
In this version of the article there are no tables and figures, which makes it impossible to get acquainted with the results of the research!
The authors need to rework the introduction - transfer the data from the "Discussion" section into it, in particular the text (lines 183-205). And some more parts of the text from this section. More clearly formulate the purpose of the study and highlight the objectives of the study.
In the section "Materials and Methods" the methodology for conducting surveys of farmers is well described. However, in the Supplementary Materials, the questionnaire questions that were asked to farmers should be presented.
This is also very interesting. The "Discussion" section should be somewhat shortened by moving some of the information to the Introduction section.
Correct the section numbering (1. Introduction; 2. Materials and Methods; 3. Results; 4. Discussion; 5. Conclusions).
Make references to cited literary sources in accordance with the rules of the journal - "This crop accounts for three quarters of the cereal production and over half the cultivated land in Niger [2] (Bezançon and Pham, 2004)".
Make a list of references in accordance with the rules of the journal.

Author Response
The article is devoted to determining the priorities of African millet breeding based on a sociological survey of farmers in the Dosso Niger region. The research carried out is very interesting.
However, the article can be published only after careful revision.
In this version of the article there are no tables and figures, which makes it impossible to get acquainted with the results of the research!
There are 6 tables and 3 figures attached
The authors need to rework the introduction - transfer the data from the "Discussion" section into it, in particular the text (lines 183-205). Done (lines 53-70) And some more parts of the text from this section. More clearly formulate the purpose of the study and highlight the objectives of the study.
In the section "Materials and Methods" the methodology for conducting surveys of farmers is well described. However, in the Supplementary Materials, the questionnaire questions that were asked to farmers should be presented.(questionnaire attached)
This is also very interesting. The "Discussion" section should be somewhat shortened by moving some of the information to the Introduction section. Done (lines 53-70)
Correct the section numbering (1. Introduction; 2. Materials and Methods; 3. Results; 4. Discussion; 5. Conclusions). Corrected line 35, 77,116, 160 and 250
Make references to cited literary sources in accordance with the rules of the journal - "This crop accounts for three quarters of the cereal production and over half the cultivated land in Niger [2] (Bezançon and Pham, 2004)". Corrected all refrences
Make a list of references in accordance with the rules of the journal. done

Reviewer 2 Report
Authors need to re-write whole manuscript. The authors must provide clear and relevant information in each section.
Author Response
Authors need to re-write whole manuscript. The authors must provide clear and relevant information in each section.
Relevant sections have been rewritten.

Reviewer 3 Report
The authors attempt to study the Breeding priorities problem for developing biofortified high-yielding pearl millet varieties and hybrids in the Dosso region of Niger. This study is interesting and has some practical value in addressing local nutritional deficiencies. The research resources are reliable and the methods are appropriate. The article is logical and well-structured. The citation is sufficient. It is a good study overall. But I want to make a few suggestions for improvement:
1. Several tables are mentioned in the results section of the article, but are not seen in the article and should be added.
2. The article should hint at the shortcomings of this study if applicable and provide an appropriate outlook for future research.
3. The article should make appropriate policy recommendations based on the findings of the study.
Author Response
The authors attempt to study the Breeding priorities problem for developing biofortified high-yielding pearl millet varieties and hybrids in the Dosso region of Niger. This study is interesting and has some practical value in addressing local nutritional deficiencies. The research resources are reliable and the methods are appropriate. The article is logical and well-structured. The citation is sufficient. It is a good study overall. But I want to make a few suggestions for improvement:
- Several tables are mentioned in the results section of the article, but are not seen in the article and should be added. There are 6 tables and 3 figures attached
- The article should hint at the shortcomings of this study if applicable and provide an appropriate outlook for future research. Done line 261-263.
- The article should make appropriate policy recommendations based on the findings of the study. Done line 263-265.

Reviewer 4 Report
Please see detailed comments in the document.

Author Response
The manuscript by Gaoh et al described an investigation taken place in Niger. The investigation is about the adoptability and acceptance of released and pipeline breeding high-iron pear millet varieties
The written of this manuscript need to be improved and the overall structure is not clear.
First, structure of the manuscripts is not clear enough. Contents in Material and Method as well as Result should be separated into different sections and start with titles. Corrected accordingly
Second, it will be nice if the author can include a brief map of Republic of Niger so that study area, departments and village locations can be outlined. Map is included
Third, although it was indicated that there are two figures and five tables in the manuscript, but I did not find any of them. Sorry if I missed them. But if not, please upload the figures and tables. There are 6 tables and 3 figures attached
Except above mentioned problems, I do have some other comments for further consideration.
Comments:
Line 13: Capital “genetic”. Corrected accordingly line 13
Line 14: Change “from 7 million ha” to “from 7 million Hectares (ha)”. Corrected accordingly line 14
Line 15: Change “Low yield is the cause of” to “Low yield is result from” Corrected accordingly line 15
Line 130-135: This sentence is too long and not clear enough to understand. Please rephrase it. Corrected accordingly line 139-141
Line 194: Change “an early variety is a only option” to “an early variety is the only option” Corrected accordingly line 171
Line 227: Change “by the both men and women farmers” to “by both men and women farmers” Corrected accordingly line 188

Round 2
Reviewer 1 Report
The authors corrected all noted remarks. The article may be published. The presented data will be of interest to both breeders and sociologists.
Author Response
Thanks to reviewe1. We have incorporated all the suggestions
Reviewer 2 Report
Comments and suggestions for authors
Title should be revised as “Establishing breeding priorities for developing biofortified high-yielding pearl millet varieties and hybrids in Dosso region of Niger”
Name and formula of statistical tests should be given in materials and methods section.
Authors have provided only frequencies and percentage data in Results section in tables and figures. They need to use statistical tests and analyzed statistical parameters in table and figure.
Author Response
Title should be revised as “Establishing breeding priorities for developing biofortified high-yielding pearl millet varieties and hybrids in Dosso region of Niger” corrected
Name and formula of statistical tests should be given in materials and methods section. It is a simple survey only ranking and percentage is used.
Authors have provided only frequencies and percentage data in the Results section in tables and figures. They need to use statistical tests and analyze statistical parameters in tables and figure. It is a simple survey only ranking and percentage is used. Similar PRA studies and papers used only percentages and ranking.

Reviewer 4 Report
I do have several minor comments as below:
Figures and Tables can be moved up to the corresponding paragraphs.
Numbers in Table 2 are better to have same decimal places. For example, 4 should be changed to 4.0.
Line 152-154: I did not find any data in figures or tables can support this statement.
Author Response
I do have several minor comments as below:
Figures and Tables can be moved up to the corresponding paragraphs. Corrected
Numbers in Table 2 are better to have same decimal places. For example, 4 should be changed to 4.0. corrected
Line 152-154: I did not find any data in figures or tables can support this statement. This statement “The FGD in the three departments revealed a big disparity between men and women regarding malnutrition. Men attributed hidden hunger to the lack of satiation, while women were more aware of micronutrient deficiencies and its consequences, mostly anemia.” was a qualitatif result of the focus group discussion with women and men separately (The FGD started with an introduction followed by mission statement to make it easier for interaction, a checklist of topics to be covered was prepared in advance and the main topics were: crops cultivated by the interviewees of the community; seasonal calendar; local perception of malnutrition; how many different varieties of millet were grown; varietal traits preferred by farmers; and pearl millet production constraints.) Statement can be removed if it is confusing?
